# Real Life Data on OnabotulinumtoxinA for Treatment of Chronic Migraine in Pediatric Age

**DOI:** 10.3390/jcm12051802

**Published:** 2023-02-23

**Authors:** Laura Papetti, Ilaria Frattale, Fabiana Ursitti, Giorgia Sforza, Gabriele Monte, Michela Ada Noris Ferilli, Samuela Tarantino, Martina Proietti Checchi, Massimiliano Valeriani

**Affiliations:** 1Developmental Neurology, Bambino Gesù Children Hospital, IRCCS, 00165 Rome, Italy; 2Child Neurology and Psychiatry Unit, Tor Vergata University of Rome, 00165 Rome, Italy; 3Center for Sensory-Motor Interaction, Aalborg University, Fredrik Bajers Vej 7 D3, DK-9220 Aalborg, Denmark

**Keywords:** OnabotulinumtoxinA, chronic migraine, adolescents, children, treatment

## Abstract

Background: The use of OnabotulinumtoxinA (OBT-A) for the treatment of chronic migraine (CM) in adults represents a therapy with the greatest efficacy and safety data. However, we have little evidence on the use of OBT-A in children or adolescents. The present study aims to describe the experience with OBT-A in the treatment of CM in adolescents in an Italian third-level headache center. Methods: The analysis included all patients under the age of 18 treated with OBT-A for CM at the Bambino Gesù Children’s Hospital. All patients received OBT-A following the PREEMPT protocol. Subjects were classified as good responders if a greater than 50% reduction in the monthly frequency of attacks was observed, partial responders if the reduction was between 30 and 50%, and non-responders if it was <30%. Results: The treated population consisted of 37 females and 9 males with a mean age of 14.7 years. Before starting OBT-A, 58.7% of the subjects had attempted prophylactic therapy with other drugs. From OBT-A initiation to the last clinical observation, the mean duration of follow-up was 17.6 ± 13.7 SD (range: 1–48) months. The number of OBT-A injections were 3.4 ± 3 SD. Sixty eight percent of the subjects responded to treatment within the first three administrations of OBT-A. Proceeding with the number of administrations, a progressive improvement in frequency was further observed. Conclusions: The use of OBT-A in pediatric age can have benefits in terms of reduction in the frequency and intensity of headache episodes. Furthermore, treatment with OBT-A has an excellent safety profile. These data support the use of OBT-A in the treatment of childhood migraine.

## 1. Introduction

Migraine is the third most frequent neurological disorder worldwide, with the second highest burden of disease [1] and the first of disability under the age of fifty [2]. Chronic migraine (CM) affects an estimated 1.5% of the global pediatric population [3,4]. Diagnosis of CM is based on the third version of the International Classification of Headache Disorders (ICHD-3), which defines CM as a condition with more than 15 headache days per month lasting more than 3 months [4]. The ultimate goals of CM therapy are to improve the quality of life and reduce the disability associated with the disease. One of the main objectives of high-frequency and chronic migraine prophylaxis is to reduce the number of attacks per month and the use of attack medications [5]. Until the advent of drugs directed against calcitonin gene-related peptide (CGRP), the prophylactic treatment of migraine used drugs originally used for other medical purposes (e.g., anti-epileptics, anti-arrhythmics, antidepressants) [6]. The intuition that OnabotulinumtoxinA (OBT-A) could be used for CM therapy arises from the description made by subjects who used it for aesthetic purposes and who simultaneously saw the improvement of the migraine trend [7]. The efficacy of OBT-A in treating migraine is due to the inhibition of nociception in the ganglia of the trigeminal neurons [8]. OBT-A acts on the peripheral nerve, interfering with SNARE-dependent exocytosis. The consequence is the blockage of fusion of synaptic vesicles on the inner face of the cell membrane and the inhibition of the release of neurotransmitters (i.e., glutamate) and neuropeptides such as CGRP, pituitary adenylate cyclase activating peptide 38 (PACAP 38), and substance P in primary sensory neurons [8,9,10]. In 2010, two trials led to the approval of the use of OnabotulinumtoxinA for the preventive treatment of chronic migraine in adults [11,12]. OBT-A treatment is still considered one of the most effective strategies for treating CM [13]. Some recent data would also suggest its use in combination with anti-CGRP antibodies to obtain greater efficacy compared to single treatments [14].

While the scientific world has collected extensive evidence of the efficacy of OBT-A on the treatment of CM in adults, we have little controlled data regarding the same in pediatric age [6,15,16,17,18,19]. Complicating the situation even more are the data related to high placebo response in children [6]. Furthermore, the possibility of clinical use of antibodies against CGRP is still far away, since pediatric clinical trials are still ongoing [20,21].

Our study aims to contribute to reporting the efficacy and safety of OBT-A in a pediatric population suffering from CM.

## 2. Materials and Methods

This study included patients with CM observed at the headache center of the Bambino Gesù Children’s hospital in Rome. The enrolment period was from November 2018 to November 2022.

The ICHD-3 criteria were used to diagnose CM as a headache occurring on 15 or more days/month for more than 3 months, which, on at least 8 days/month, has the features of migraine headache.

Patients received OBT-A at doses of 155–195 units according to the Phase 3 Research Evaluating Migraine Prophylaxis Therapy (PREEMPT) protocol [11]. OBT-A therapy was proposed for patients after the failure of at least two traditional drug therapies, and it was administered after obtaining written informed consent from the parents.

The criteria for CM were verified in the three months prior to the initiation of OBT-A therapy through the administration of an attack diary in order to collect the monthly headache days (MHDs). The diary was also used to assess the frequency and intensity of the headache episodes over time to OBT-A treatment. The baseline was defined as the monthly mean of the 3 months preceding OBT-A treatment. Concomitant use of other medications for migraine prophylaxis was permitted.

The primary endpoint was to observe changes in MHDs from baseline during the three cycles of OBT-A treatment (Cy1–3). Patients were defined as follows according to their response to OBT-A: 1. good responders (GR) if achieving a ≥50% reduction in MHDs, 2. partial responders (PR) if achieving a 30–49% reduction in MHDs, and 3. non-responders (NR) if achieving a <30% reduction in MHDs from baseline to the respective 3-month cycles. The NR group also included patients lost to follow-up or discontinuing treatment before the third cycle.

The secondary endpoints included the assessment of variation in intensity of the attacks. Each subject was asked to indicate in the diary, before and during treatment with OBT-A, the intensity of the attacks using a score from 1 to 3 (1 mild, 2 moderate, and 3 severe). The average of the scores obtained was then considered to distribute the patients into three categories: mild attacks (average score from 0 to 1); moderate attacks (average score between 1 and 2), and severe attacks (average score between 2 and 3).

Finally, within 3 months before the start of OBT-A, the patients were offered, with their consent, the administration of psychological tests to evaluate the presence of anxiety and depression. Anxiety was evaluated using the General Anxiety Disorder-7 (GAD-7) scale; depression was estimated using the Patient Health Questionnaire-9 (PHQ-9) scale. The GAD-7 is a brief, 7-item, self-reported measure of anxiety, rated on a 4-point Likert scale. Four alternatives were offered: 1: not at all; 2: some days; 3: more than half the days; and 4: almost every day. The total score was categorized into “no anxiety” (range: 0–4), “mild anxiety” (range: 5–9), “moderate anxiety” (range: 10–14), and “severe anxiety” (equal to 15) [22]. The PHQ-9 is a brief, self-administered measure of depressive symptoms, with nine items that fit the diagnostic criteria for major depressive disorder. Item 9 investigates the idea of harming oneself or wanting to die. This is a screening instrument and, thus, does not represent a clinical diagnosis. Symptoms are rated using a 4-point scale (0 = not at all; 1 = some days; 2 = more than half the days; and 3 = almost every day) regarding the past two weeks experienced. The severity of depressive symptoms was categorized as “no depression” (range: 0–4), “mild depression” (range: 5–9), “moderate depression” (range: 10–14), and “severe depression” (equal to 15) [23,24].

The analysis was performed using SPSS version 27 software. The dependent samples *t*-test was used to analyze the changes in the averages of MHDs and the intensity of attacks over time (OBT-A cycles). We compared the three groups of GR, PR, and NR using the Mann–Whitney U-tests for ordinal categorical variables (sex, intensity of the attacks, presence or absence of anxiety and depression) and the independent *t*-test (mean and standard deviation (SD)) for ordinal variables (frequency of attacks and age). Statistical significance was set as *p* < 0.05.

The study was conducted in accordance with the Declaration of Helsinki and approved by the Institutional Review Board of Bambino Gesù Children’s Hospital (protocol code not applicable). Informed consent was obtained from all subjects involved in the study.

## 3. Results

In the considered period, 179 patients under 18 years of age were observed with the diagnosis of CM. Of these, 46 patients underwent at least one administration of OBT-A (Botox^®^). The general characteristics of the population and migraine history and previous treatments are summarized in Table 1.

The treated population consisted of 37 females and 9 males with a mean age of 14.7 ± 1.5 standard deviation (SD) years (range: 12–17). Patients had a mean disease duration from onset to initiation of OBT-A of 29.3 ± 9.1 SD months. The average weight of patients receiving treatment was 41.3 ± 5.1 kg.

Before starting OBT-A, all patients had failed at least two treatments with a previous drug (Figure 1). None had improvements in efficacy. Furthermore, side effects were present in 58.7% of cases treated with traditional drugs. All the side effects were attributable to the known safety profile of the molecules.

Of the total of 46 subjects, 28 patients (60.8%) started treatment with OBT-A concomitantly with the administration of another drug (Figure 2).

The mean time on drug therapy initiation before the first injection of OBT-A was 2.1 months, while the mean time off after starting OBT-A was 3 months.

Of the 46 subjects who were administered at least one injection, 9 (19.5%) stopped the treatment before completing the 3 doses reporting that they could not tolerate the injections because they were painful. However, three of these subjects had received a partial response. Those who stopped their cycles early were younger than those who tolerated the treatment (mean age 13.67 ± 1.5 vs. 15.1 ± 1.4 years; *p* < 0.01).

Three patients who received a first dose of OBT-A were excluded from the efficacy analysis because, at the time of writing the paper, they had a follow-up of less than 1 month from the first injection. Therefore, the efficacy analysis was performed on 43 patients. In Figure 3, we have summarized the percentages of NR, PR, and GR at each cycle. NR patients also included those who discontinued treatment because they could not tolerate the injections. Sixty eight percent of the subjects achieved a response (GR and PR) to treatment within the first three administrations of OBT-A. A response to treatment was observed in 35% of subjects (28% PR and 7% GR) at the first administration; at the second administration, there were 58% of responders (32% PR and 26% GR). From the third administration, the percentage of NR remained stable, while in the responder population, there was an increase in patients whose response improved, passing from PR to GR groups (Figure 3).

MHDs progressively decreased from baseline (mean 21.8 ± 4.8 SD) to Cy1 (mean 16.6 ± 5.5 SD; −5.25; *p* < 0.0001), from Cy1 to Cy2 (mean 12.3 ± 6.3; −4.3; *p* < 0.0001), from Cy2 to Cy3 (mean 10.2 ± 6.7 SD; −2.07; *p* < 0.001), and from Cy3 to Cy4 (mean 9.5 ± 6.8 SD; −0.67; *p* < 0.05 (Figure 4).

Figure 5 shows that as treatment with OBT-A continues at each cycle, there is an increase in subjects with an average score for the intensity which can be classified as mild, while the number of severe attacks progressively decreases.

Sixty six percent of non-responders and 54% of responders underwent concomitant treatment with another drug, with no differences between the two groups (*p* < 0.05).

Side effects were observed in 32% of subjects, but no patients experienced serious adverse effects. The most frequently reported situations were injection site edema (5%) and pruritus (4%); headache (5%); neck muscle weakness (1%); and neck pain (1%).

Of the 43 patients included, 37 were administered GAD-7 and PHQ-9 tests. In the comparison between responders and NR to OBT-A treatment, we found no significant differences between the scores of the psychological tests (Table 2).

For demographic characteristics, we found that NR subjects were more frequently female than male (40% vs. 60%; *p* < 0.05).

## 4. Discussion

In this study, we observed that treatment with OBT-A is effective in the majority of treated subjects, suggesting that the use of OBT-A can also be a valid therapeutic strategy in children and adolescents affected by CM.

A significant reduction in the frequency of MHDs is obtained for the majority of patients (68%) within the first three cycles of OBT-A treatment. At the end of the first three administrations, 45% of the population studied had a reduction in MHDs greater than 50%.

Continuing with the cycles beyond the third, a trend towards the stability of the percentage of NR subjects was observed, while in the responder population, a trend towards further improvement in the frequency of the headache attacks was observed with an increase in the number of GR subjects compared to PR.

Failure to adhere to treatment due to low tolerance to injections was observed in 19.5% of cases, suggesting that the treatment is generally well tolerated. A major difficulty was observed in younger pre-adolescent children who are the ones who most frequently interrupt treatment early.

The side effects reported in 32% of patients are minor and temporary, while none of the treated patients experienced severe complications. Our treatment-related adverse effects were consistent with the known tolerability profile of OBT-A when injected into the head and neck muscles [25]. No new safety concerns were discovered. This underlines how treatment with OBT-A also has an excellent safety profile in pediatric age.

When comparing responders and NR patients, we found no significant differences in baseline migraine characteristics and psychological test scores for anxiety and depression. The fact that NR were more frequently female is probably a bias resulting from the greater number of female subjects enrolled than males.

Overall, the results of our work suggest that the use of OBT-A can be considered among the treatment strategies of CM in the pediatric age.

OBT-A is an approved therapy for the treatment of CM in adulthood. This result was achieved through the publication of RCT [11,12].

In the literature, there are little data on the use of OBT-A in children. These data mostly come from studies on a limited number of subjects and with methods of administration of OBT-A according to heterogeneous doses and schedules [15,16,17,18,19]. Collectively, these studies show efficacy and safety data of OBT-A for pediatric chronic migraine [15,16,17,18].

In detail, we have only two controlled studies on the use of OBT-A in children and adolescents [18,19]. One RCT involved subjects with CM aged between 8 and 15 years. In this study, OBT-A was administered at a dosage of 155 units at 31 injection sites, with cycles repeated every 3 months. Attack frequency trends were evaluated with follow-up visits every 6 weeks. Subjects that received OBT-A (n° 9) reported a statistically significant decrease from the baseline compared with placebo (n° 6) in monthly frequency and (*p* < 0.05) intensity of the headache attacks (*p* < 0.05) and PedMIDAS (*p* < 0.05) [18].

Another RCT involved 115 patients aged 12 to 17 years and assessed a single treatment of OBT-A (155 U or 74 U) vs. placebo (intramuscular saline) administered according to the PREEMPT approach. The primary efficacy measure was change in the frequency of headache days from baseline to week 12. All treatments reduced frequency of headache days at week 12, with no significant differences between treatments (−6.3 for OBT-A 155 U; −6.4 for 75 U and −6.8 for placebo, *p* > 0.47). In all groups, the frequency of severe headache days was reduced, and treatments were well tolerated [19]. Although this study did not demonstrate the achievement of the efficacy endpoints for OBT-A, definitive conclusions cannot be drawn, as it considers the effects obtained after only one course of therapy. This is a very strong limitation especially with regard to the estimation of the response to the placebo group.

There is a large amount of data on adults showing that many patients with CM treated with OBT-A who did not respond to the first treatment cycle responded in the second and third cycles of treatment [26]. Furthermore, the PREEMPT extension studies document that the long-term use of OBT-A (after the third administration) with cycles repeated every 3 months leads to further improvements in terms of monthly frequency of attacks while also maintaining an excellent safety profile [27,28].

Our study has the limitation of not having a placebo control group. Unfortunately, this limitation is difficult to overcome because, for ethical reasons, it becomes difficult to justify long-term treatment with placebo in injection mode in children. Compared to previously published studies, ours involves a large pediatric population and evaluates the use of OBT-A following the PREEMPT protocol without changes based on age or weight. Furthermore, we evaluated the efficacy and safety of OBT-A after at least three treatment cycles and not only at the first administration.

Some studies on adults and children have tried to understand if there could be predictive factors of response or lack of response to treatment with OBT-A [15,29,30,31,32]. In adults with CM, the factors associated with a better response to treatment with OBT-A would be male sex [29] and the presence of MOH at baseline [31], while it is not clear whether the high number of MHDs at baseline could be a negative or positive predictive factor of response to OBT-A [31,32].

A study conducted in pediatric-age CM patients attempted to analyze possible clinical data predictors of response to therapy with OBT-A [15]. The authors found that in the 34 subjects treated with OBT-A, there was a significant reduction in the MHDs (*p* < 0.001) and the intensity of attacks (*p* < 0.005). In NR, 67% of subjects had significant scores for an anxiety disorder identified by GAD-7 tests (vs. 24% of responders, *p* < 0.05). The authors therefore concluded that in pediatric CM, comorbidity with an anxiety disorder may be a predictor of poor clinical response to treatment [15]. In our cohort, we tried to understand if there could be a difference in GAD-7 scores for anxiety and PHD-9 scores for depression between responders and NR. However, although the majority of total patients presented with an anxiety disorder (69.5%) or depression (65%), we did not find significant differences in relation to response or no response to treatment with OBT-A.

However, the high prevalence of anxiety and depression in pediatric patients with CM and the possibility that these disorders could influence the response to a specific treatment reaffirm the importance of managing the psychological aspects as an integral part of the treatment strategy for migraine in developmental age.

Our study has the limitations of lack of a placebo control group and the small number of patients. However, we think it can help to provide the cue to consider OBT-A in the treatment of CM also in young subjects. This is very important because in the pediatric age, it is not possible to overcome many methodological limits linked to the realization of controlled studies and, from a practical point of view, the choice of drugs is based on conflicting data present in the literature and experiences coming from the treatment of migraine in ‘adults’. Furthermore, while awaiting the results of the pediatric trials on anti-CGRPs, we think it may be useful to expand the data on other available therapies.

## 5. Conclusions

Our study supports the fact that OBT-A can be used for the treatment of CM in children and adolescents. OBT-A has been shown to be effective in reducing the number of days with headache and the intensity of attacks in our population. No patient developed major adverse effects. Response to OBT-A may be assessed over a minimum of 2–3 cycles (6–9 months). In responding patients, long-term treatment beyond the third cycle can lead to further benefits in terms of reducing the frequency and intensity of headache attacks. We found no predictive factors to distinguish responders from non-responders. Further studies with controlled data should support the data of the efficacy of multiple treatment cycles of OnabotulinumtoxinA for CM prevention in pediatric age.

## Figures and Tables

**Figure 1 jcm-12-01802-f001:**
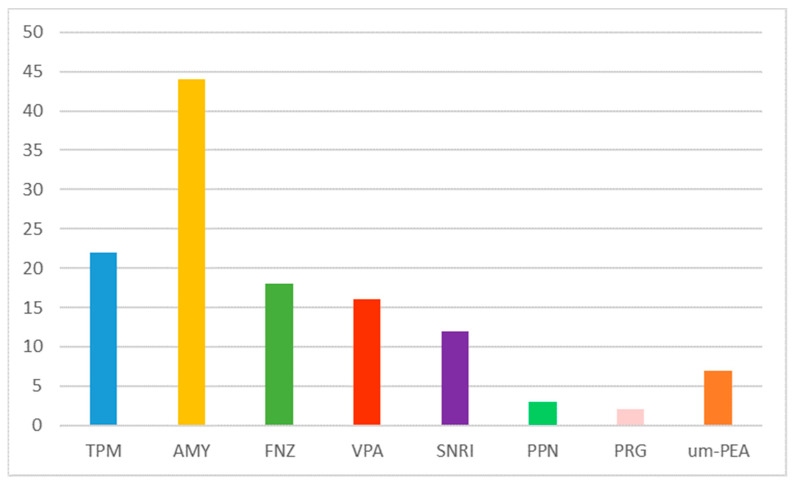
Prophylactic medications taken before starting OBT-A (frequency). AMY: amitriptyline; VPA: valproate; SNRI: serotonin-norepinephrine reuptake inhibitor; PPN: propranolol; LMT: lamotrigine; FNZ: flunarizine.

**Figure 2 jcm-12-01802-f002:**
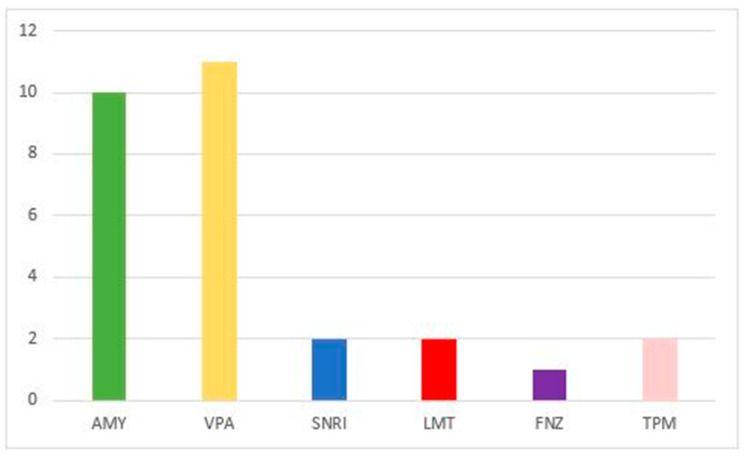
Drugs used concurrently with OBT-A initiation (frequency).

**Figure 3 jcm-12-01802-f003:**
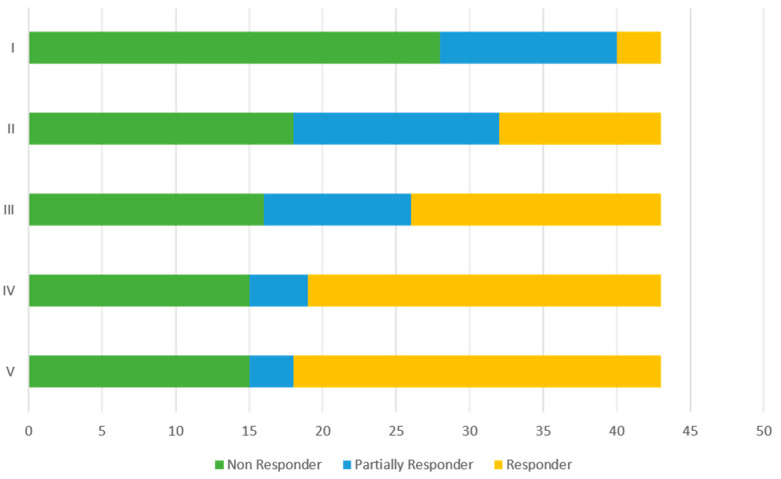
Frequencies of non-responders (NR), partial responders (PR), and good responders (GR) at each OBT-A cycle.

**Figure 4 jcm-12-01802-f004:**
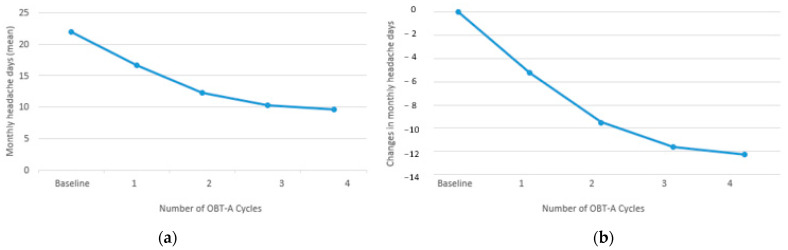
Changes in monthly headache days (MHDs) from baseline along with OBT-A cycles: (**a**) number of the attacks for month (mean); (**b**) difference between baseline and each cycle.

**Figure 5 jcm-12-01802-f005:**
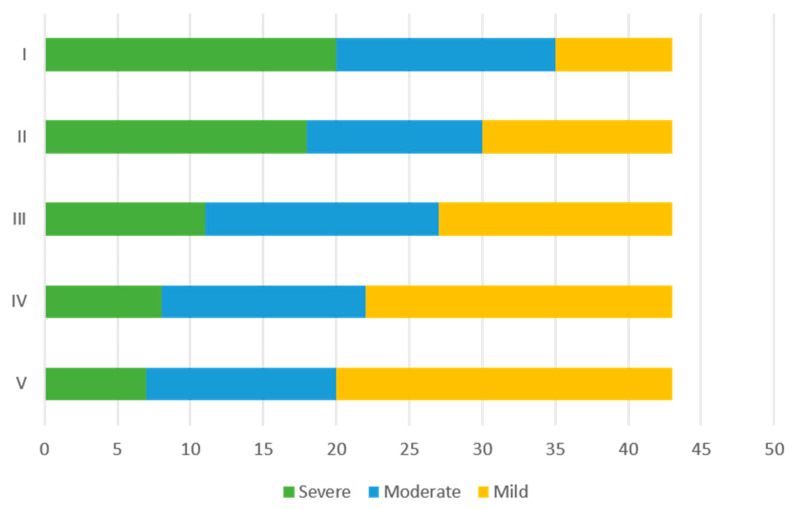
Frequencies of patients reporting an average of mild, moderate, or severe intensity of headache attacks for each cycle of OBT-A.

**Table 1 jcm-12-01802-t001:** Features of 46 patients receiving just a dose of OBT-A. TPM: topiramate; AMY: amitriptyline; FNZ: flunarizine; VPA: valproate; SNRI: serotonin-norepinephrine reuptake inhibitor; PPN: propranolol; PRG: pregabalin; um-PEA: ultra-micronized palmitoylethanolamide.

Age (Years)	Mean 14.7 ± 1.5 SD Range: 12–17
Sex	9 males (19.6%) 37 females (80.4%)
Duration of migraine history (months) Concomitant medication overuse headache (MOH)	Mean 29.3 ± 9.1 SD Range: 8–36 7 subjects (15%)
Previous prophylactic treatment n° (%)	Amitriptyline 44 (95.7%) Topiramate 22 (47.8%) Flunarizine 18 (39.1%) Valproate 16 (34.8%) Duloxetine 12 (26.1%) Propanolol 3 (6.5%) Pregabalin 2 (4.3%) Ultra-micronized palmitoylethanolamide (um-PEA) 7 (15.2%) Cognitive behavioral therapy 30 (65.2%)
Day with headache for month	Mean: 26.5 ± 5.8 SD Range: 15–31
Number of inoculums (mean + SD; range)	Mean: 3.4 ± 3 SD Range: 1–17
Associated drug n° (%)	None 18 (39.1%) Amitriptyline 10 (21.7%) Valproate 11 (23.9%) Duloxetine 2 (4.3%) Lamotrigine 2 (4.3%) Flunarizine 1 (2.2%) Topiramate 1 (2.2%)

**Table 2 jcm-12-01802-t002:** Differences in psychological outcomes between responders and non-responders.

Patients Performed Tests (n° 37)	Total	Non-Responder (n° 14)	Responder (n° 23)	Sig. *p*
No Depression	8	4 (28.6%)	4 (17.4%)	0.343
Mild Depression	11	3 (21.4%)	8 (34.8%)	0.316
Moderate Depression	10	4 (28.6%)	6 (26.1%)	0.58
Severe Depression	8	3 (21.4%)	5 (21.7%)	0.657
No Anxiety	7	3 (21.4%)	4 (17.4%)	0.541
Mild Anxiety	10	2 (14.3%)	8 (34.8%)	0.164
Moderate Anxiety	11	5 (35.7%)	6 (26.1%)	0.397
Severe Anxiety	9	4 (28.6%)	5 (21.7%)	0.464

## Data Availability

Data available on request due to privacy restrictions.

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
