# Peer review of "Real Life Data on OnabotulinumtoxinA for Treatment of Chronic Migraine in Pediatric Age"

_jcm, 2023, doi:10.3390/jcm12051802_

Round 1

Reviewer 1 Report

The authors present an open-label uncontrolled study of the use of onabotulinumtoxinA in the pediatric age group. The study appears to be well done, with interesting findings.

I request that the authors consider the following comments in order to improve the clarity of their paper.

1.       In the introduction it is stated that: “Chronic migraine (CM) affects 9% of the world's pediatric population [3].” This seems very high.  In the general population, the prevalence of chronic migraine is usually considered to be about 1 – 2%.  Could the authors check their statement with some additional references? Perhaps their figure applies to a clinic population?

2.      In the methods it is stated that  “OBT-A therapy was proposed for patients after failure of at least two traditional drug therapies.”  Yet in the abstract and elsewhere, it is stated that “Before starting the OBT-A, 58.7% of the subjects had attempted prophylactic therapy with other drugs.”  The two statements seem to contradict each other.  Can the authors please clarify?

3.      I don’t understand figure 3.  Should not the sum of the percentages for responders, partial responders, and non-responder equal 100%?   It seems to equal about 43 % in the figure.  

4.      Same for figure 5.  Should not the sum of mild, moderate, and severe attacks equal 100%?

Author Response

1- 9% is the prevalence of chronic migraine in the pediatric migraine population. Between 1 and 2% the prevalence in the global pediatric population. We changed the sentence.

2. sorry for the misunderstanding. In the sentence reported, the value 58.7% refers to the appearance of side effects with traditional pharmaceuticals. we have rewritten the sentence as follows

"Before starting the OBT-A, all patients had failed at least two treatments with a previous drug (figure 1). None had improvements in efficacy. Furthermore, side effects were present in 58.7% of cases treated with traditional drugs. All the side effects were attributable to the known safety profile of the molecules"

3 and 4. both figure 4 and figure 5 have an error in the legend. We have not reported the percentages but the frequencies. The legends have therefore been corrected.

Reviewer 2 Report

1. According to "Materials and Methods" OBT-A was proposed for patients after failure of at least two traditional drug therapies. But in "Results" only 58.7% of the subjects had attempted a previous prophylaxis therapy with traditional drugs. 

2. Information about other bodersome symptoms (nausea, vomiting, photophobia, etc) may be interesting to assess OBT-A efficacy. 

Author Response

1. 

sorry for the misunderstanding. In the sentence reported, the value 58.7% refers to the appearance of side effects with traditional pharmaceuticals. we have rewritten the sentence as follows

"Before starting the OBT-A, all patients had failed at least two treatments with a previous drug (figure 1). None had improvements in efficacy. Furthermore, side effects were present in 58.7% of cases treated with traditional drugs. All the side effects were attributable to the known safety profile of the molecules"

2. sorry but changes in accompanying symptoms (nausea, photophobia etc) were not among the response endpoints.
therefore the variation was not considered. In the methods, the measure of response to treatment was specified (effectiveness in terms of reduction and intensity of attacks).

Reviewer 3 Report

I read with the great  interrest your article. It is very interresting topic. The article should be accepted as written.   

Author Response

Thanks for considering this paper for publication.